# Peptide barcode of multidrug-resistant strains of *Neisseria gonorrhoeae* isolated from patients in Thailand

Sittiruk Roytrakul[1], Pongsathorn Sangprasert[2], Janthima Jaresitthikunchai[1], Narumon Phaonakrop[1], Teerakul Arpornsuwan[3]*

1 Functional Proteomics Technology Laboratory, National Center for Genetic Engineering and Biotechnology, Khlong Luang, Pathumthani, Thailand, 2 Graduate Student of Department of Medical Technology, Faculty of Allied Health Sciences, Thammasat University, Khlong Luang, Pathumthani, Thailand, 3 Medical Technology Research and Service Unit, Health Care Service Center, Faculty of Allied Health Sciences, Thammasat University, Khlong Luang, Pathumthani, Thailand

* teerakul@tu.ac.th

## Abstract

The emergence of multidrug-resistant strains of *Neisseria gonorrhoeae* constitutes a serious threat to public health. The present study aimed to investigate peptidome-based biomarkers of multidrug-resistant *N. gonorrhoeae*, using matrix-assisted laser desorption/ionization time-of-flight mass spectrometry (MALDI-TOF MS) and liquid chromatography tandem mass spectrometry (LC-MS). The peptide barcode database of multidrug resistant *N. gonorrhoeae* was generated from the whole-cell peptides of 93 *N. gonorrhoeae* isolated from patients in Thailand. The dendrogram of 93 independent isolates of antibiotic-resistant *N. gonorrhoeae* revealed five distinct clusters including azithromycin resistance group (AZ), ciprofloxacin resistance group (C), ciprofloxacin and penicillin resistance group (CP), ciprofloxacin and tetracycline resistance group (CT), ciprofloxacin, penicillin and tetracycline resistance group (CPT). The peptidomes of all clusters were comparatively analyzed using a high-performance liquid chromatography-mass spectrometry method (LC-MS). Nine peptides derived from 9 proteins were highly expressed in AZ ($p$ value < 0.05). These peptides also played a crucial role in numerous pathways and showed a strong relationship with the antibiotic resistances. In conclusion, this study showed a rapid screening of antibiotic-resistant *N. gonorrhoeae* using MALDI-TOF MS. Additionally, potential specific peptidome-based biomarker candidates for AZ, C, CP, CT and CPT-resistant *N. gonorrhoeae* were identified.

## 1. Introduction

*Neisseria gonorrhoeae* is the strict human pathogenic bacterium that causes the sexually transmitted infection which usually affects the mucous membranes of the urethra in males and the endocervix and urethra in females. Although the prevalence of the infection is low but it can have serious subsequent complications. In 2020, it was estimated by WHO that there were 82.4

**Funding:** The author(s) received no specific funding for this work.

**Competing interests:** The authors have declared that no competing interests exist.

million newly infected cases among adolescents and adults aged 15–49 years worldwide with a global incident rate of 19 (range of 11–29) per 1,000 women and 23 (range of 10–43) per 1,000 men. In Thailand, the incidence of multidrug resistant *N. gonorrhoeae* isolates was found to be 10.4, 10.5 and 13.1 per 100,000 population in 2013–2015, respectively [1, 2]. The spread of multidrug resistant *N. gonorrhoeae* tends to be a serious problem for the treatment and control of gonorrhea worldwide [3]. Treatment failure to cephalosporin and/or elevated MIC to ceftriaxone (MIC $\geq$ 0.25 µg/ml) and widespread emergence of cephalosporin resistant *N. gonorrhoeae* infection in Western Pacific and South East Asia region countries have been reported [4]. Currently, there was no effective vaccine against gonococcal infections. Furthermore, antimicrobial resistances to the drugs previously used for its treatment, including sulfonamides, penicillin, narrow-spectrum cephalosporins, tetracyclines, macrolides, and fluoroquinolones have been developed [5].

In a study of the US Gonococcal Isolate Surveillance Project, *N. gonorrhoeae* isolates with decreased susceptibility to azithromycin (minimal inhibitory concentration [MIC] $\geq$ 2 µg/mL) were whole-genome sequenced. The genomic diversity, strain population dynamics, and antimicrobial resistance profiles of *N. gonorrhoeae* isolates were established by using Bioinformatic analyses. The study showed that decreased azithromycin susceptibility was involved in expanding and persistent clades harboring two well described resistance mechanisms, mosaic mtr locus and 23S rRNA variants [6]. The global protein expression under ESC stresses of ESC-susceptible and ESC-reduced susceptible *N. gonorrhoeae* strains were demonstrated by using two-dimensional gel electrophoresis and MALDI-TOF/TOF-MS analysis. The results showed the expression of several proteins implicated in a variety of biological functions including transport system, energy metabolism, stress response and pathogenic virulence factors for ESC antibiotics. The increased expression of ESC-reduced susceptible strain under ESC stress was shown to be macrophage infectivity potentiator (Ng-MIP) [7].

Up to date, the detection and typing of various microorganisms using by Matrix Assisted Laser Desorption Ionization Time of Flight Mass Spectrometer (MALDI-TOF MS) is considered as accurate, reliable, economic, and rapid tool for the workflow of microbiology laboratories [8]. MALDI-TOF MS was successfully used for identification of 92 clinical isolates from 93 gonococcal isolates collected from 2007 to 2012 as part of the European Gonococcal Antimicrobial Surveillance Program [9]. Arpornsuwan et al. (2014) reported that 18 antibiotic resistant and sensitive strains of *N. gonorrhoeae* were discriminated by using MALDI-TOF MS. However, MALDI Biotyper could not distinguish between clinical isolates of *N. gonorrhoeae* that were resistant to 3 antibiotics (ciprofloxacin, penicillin and tetracycline) from those resistant to ciprofloxacin and penicillin or ciprofloxacin and tetracycline [10]. Novel peptidome biomarkers can provide strong supporting evidence for diagnosis, prognosis, monitoring, and selecting treatment options. The purpose of this study was to investigate the peptidome-based biomarkers of 93 *N. gonorrhoeae* isolates in relation to their MIC of azithromycin, ceftriaxone, cefixime, ciprofloxacin, penicillin and tetracycline. The MALDI-TOF MS spectra were visually analyzed to assess the presence of distinctive peak(s). LC-MS was used to quantify and identify the potential peptide biomarkers. Bioinformatic analyses established the peptide diversity and the relationship between antibiotic resistant gonococcal isolates with the distinct peptides and antibiotic susceptibility.

## 2. Materials and methods

### 2.1 Bacterial strains

From January 2019 to December 2021, a total of 93 antibiotic-resistant *N. gonorrhoeae* strains were isolated from urethra, rectum, pharynx and cervix of male and female patients in

Thailand by the Sexually Transmitted Infection Cluster, Bureau of AIDS TB and STIs, Department of Disease Control, Ministry of Public Health, Thailand. The experimental research was conducted in November 2022. The study protocol was approved by the Human Research Ethics Committee of Thammasat University (Science), Thailand (August 8, 2022) (Project No. 071/2565) and was performed in line with the principles of the Declaration of Helsinki, the Belmont Report, CIOMS Guidelines and the International Practice (ICH-GCP). All data were fully anonymized before we accessed them and the Institutional Review Board (IRB) or Ethics Committee waived the requirement for informed consent.

All 93 gonococcal strains were resistant to tetracycline and 31 isolates were penicillinase-producing *Neisseria* gonorrhoeae (PPNG) negative and 62 isolates were PPNG positive. There were 82 isolates which were resistant to ciprofloxacin but susceptible to cefixime, azithromycin and ceftriaxone. Only 11 isolates were susceptible to cefixime, ciprofloxacin, ceftriaxone and azithromycin. *N. gonorrhoeae* ATCC 49226 which was susceptible to azithromycin, *N. gonorrhoeae* 17GQA05 and GCREF2012011 which were resistant to azithromycin and *N. gonorrhoeae* 18GQA02, WHO K, WHO L, 01G0232 which were resistant to ceftriaxone and cefixime were used as the reference stains.

## 2.2 Bacterial culture conditions

All drug resistant *N. gonorrhoeae* strains were cultured overnight on GC agar base (Difco BD Diagnostic Systems, USA) supplemented with IsoVitaleX Enrichment (BD Diagnostic Systems, USA) at 35–36˚C in 5% $CO_2$. Before testing, the bacterial strains were cultured to reach logarithmic phase of growth in GC broth with the same supplements overnight at 35–36˚C in 5% $CO_2$.

## 2.3 Antimicrobial susceptibility testing

Determinations of antimicrobial susceptibility of *N. gonorrhoeae* isolates and their minimum inhibitory concentrations (MICs) of azithromycin (AZ), ceftriaxone (TXE), cefixime (IXE), tetracyclines (TCE), ciprofloxacin (CIPE), and penicillin (PC) were performed by using the GC agar base supplemented with IsoVitaleX Enrichment. The MICs were determined by the E-test method in accordance with the guidelines of the Clinical Laboratory Standards Institute (CLSI document m100-s25). The E-test strip evaluates the inhibitory potential of a single antimicrobial agent over a large range of concentrations (e.g., 0.032–256 μg/mL). The MICs were interpreted by reading the scale in terms of μg/mL where the ellipse edge intersects the strip. After the required incubation period, the MIC value is read where the edge of the inhibition ellipse intersects the side of the strip. The presence of beta-lactamase enzyme production and resistance to penicillin were detected by using the nitrocefin test (Becton, Dickinson and Company, Thailand). The penicillinase-producing *N. gonorrhoeae* (PPNG) was defined as isolates testing positive for beta-lactamase enzyme. The working stock of *N. gonorrhoeae* isolates was stored at -80˚C. *N. gonorrhoeae* ATCC 49226 and WHO reference strains K and L were used as quality control in each assay for validation.

## 2.4 Peptide barcoding by MALDI-TOF MS analysis

The bacterial whole cells were transferred into 900 μl of sterile distilled water, suspended, and mixed with 300 μl of absolute ethanol. After centrifugation, the pellet was resuspended and vigorously mixed with MALDI matrix solution (10 mg sinapinic acid in 1 mL of 50% acetonitrile containing 2.5% trifluoroacetic acid). Two microliters of the peptidomes were directly spotted onto MALDI target (MTP 384 ground steel, Bruker Daltonik, GmbH) and allowed to dry at room temperature. MALDI-TOF MS spectra were collected using Ultraflex III TOF/

TOF (Bruker Daltonik, GmbH) in a linear positive mode with mass range of 2,000–20,000 Da. Five hundred shots were accumulated with a 50 Hz laser for each sample. MS spectra were analyzed by using FlexAnalysis and ClinProTool software (Bruker Daltonik, GmbH). Three major statistical tests, consisting of Anderson-Darling (AD), t-test/ANOVA (TTA), and Wilcoxon/ Krustal-Wallis (W/KW) tests, were incorporated into ClinProTools. Data with a normal distribution are subjected to TTA test, while those of a non-normal distribution subjected to W/ KW test and AD test determines whether test data are based on normal distribution assumption (considering *p*-value of $> 0.05$ for normal distribution and of $\leq 0.05$ for non-normal distribution). The unsupervised hierarchical clustering (dendrogram) was used to examine the clustering of all isolates. ACTH fragment 18–39 (human), insulin oxidized B chain (bovine), insulin (bovine), cytochrome C (equine) and apomyoglobin (equine) were used as external protein calibrations.

## 2.5 LC-MS/MS analysis

The bacterial peptides were extracted from the *N. gonorrhoeae* according to a previous study with slightly modification [11]. Briefly, 5% (v/v) trifluoroacetic acid (TFA) in absolute acetonitrile (ACN) was added to the culture and the suspension was dissolved by gentle vortexing. The samples were dried and resuspended with 0.1% (v/v) formic acid. The Lowry assay was used to determine peptide concentration [12]. Peptides were equally pooled from each bacterium within the same group. Peptidome samples of each bacterial group including azithromycin resistance group (AZ), ciprofloxacin resistance group (C), ciprofloxacin and penicillin resistance group (CP), ciprofloxacin and tetracycline resistance group (CT), ciprofloxacin, penicillin and tetracycline resistance group (CPT) were analyzed by LC-MS. Briefly, 5 μg of pooled samples were reduced with 10 mM DTT and alkylated with 100 mM iodoacetamide. After trypsin digestion overnight at 37˚C, peptides were submitted to a gradient-eluted peptides using an Ultimate 3000 LC System coupled to an HCTUltra PTM Discovery System (Bruker Daltonics, Bremen, Germany). Peptides were separated on a nanocolumn (PepSwift monolithic column 100 μm i.d. x 50 mm). Eluent A was 0.1% formic acid and eluent B was 80% acetonitrile in water containing 0.1% formic acid. Peptide separation was achieved with a linear gradient from 10% to 70% B for 13 min at a flow rate of 300 nL/min, including a regeneration step at 90% B and an equilibration step at 10% B, one run took 20 min. Peptide fragment mass spectra were acquired in data-dependent AutoMS mode with a scan range of 300 −1,500 m/z, 3 averages, and up to 5 precursor ions selected from the MS scan 50−3,000 m/z. The LC-MS analysis of each sample was done in triplicate.

## 2.6 Peptidomics data analysis

The DecyderMS software was used to quantify the peptides in individual samples from MS/ MS data [13]. It provides novel 2D and 3D visualizations of LC-MS data to allow for raw data quality assessment and interactive confirmation of results achieved using automated methods for peptide detection, charge state assignments, and peptide matching across multiple LC-MS experiments. Univariate statistical tools such as Students t test and ANOVA are available to identify significantly varying peptides among different groups of samples. The Mascot search engine was used to correlate MS/MS spectra obtained from DecyderMS software to the NCBI *Neisseria gonorrhoeae* database [14]. The parameters for the database search were taxonomy (*Neisseria gonorrhoeae*), enzyme (NoCleave), variable modifications (oxidation of methionine residues), mass values (monoisotopic), protein mass (unrestricted), peptide mass tolerance (1.2 Da), fragment mass tolerance (±0.6 Da), peptide charge state (1+, 2+ and 3+), and missed cleavages (3). Proteins considered as identified proteins had at least one peptide with an

individual mascot score corresponding to $p < 0.05$. The peptidomic heatmap, principal component analysis (PCA) and analysis of variance (ANOVA) analysis were generated using Metaboanalyst 5.0 [15]. All differentially expressed peptides were analyzed for their intersections among the different sample groups using jvenn [16]. Information about particular proteins was used in the annotation by UniProtKB/Swiss-Prot entries (**http://www.uniprot.org/**). The relationships between identified proteins (from *N. gonorrhoeae*) and antibiotics were investigated using STITCH 5.0 [17].

### 2.7. Data analyses

Statistical analysis was performed using GraphPad Prism Version 6.0 for Windows (GraphPad Software, San Diego, CA, USA) and Microsoft Excel 2013 for all data analyses and graphs. *P* values of less than 0.05 ($p < 0.05$) were considered statistically significant.

## 3. Results

### 3.1 Demographic characteristics of clinical isolates

Overall, 47% of *N. gonorrhoeae* clinical isolates tested for antimicrobial susceptibility in this study were from men, 53% were from women. Thirty-nine, 37, 9 and 8 bacteriral strains were isolated from urethra, endocervix, rectum and pharynx of the patients, respectively. Among 39 isolates from urethra, 31 were from male and 8 were from female patients. Most positive specimens from men were from urethral (70.5%) rectal (20.5%) and pharyngeal swabs (9%) while those from women were predominately from endocervical (75.5%), urethral (16.3%) and pharyngeal swabs (8.2%).

### 3.2 Antimicrobial susceptibility testing

Antimicrobial susceptibility testing of 93 multidrug resistant *N. gonorrhoeae* clinical isolates were performed by using E-test and phenotypic confirmation of extended-spectrum-lactamase using BCP acidometric method. The antibiotic susceptibility test showed that among the 93 clinical isolates which were resistant to penicillin, 62 strains (66.7%) were positive and 31 strains (33.3%) were negative for beta-lactamase. The positive beta-lactamase was referred as PPNG strain, and those negative for beta-lactamase was referred as non-PPNG strain. The 93 antibiotic-resistant *N. gonorrhoeae* clinical isolates were divided into five groups: 21 isolates of the first group were non susceptible to azithromycin and intermediate resistance to tetracycline, penicillin and negative for beta lactamase, but susceptible to cefixime, ciprofloxacin and ceftriaxone, only one isolate was non susceptible to azithromycin and intermediate resistance to tetracycline and negative for beta lactamase, but susceptible to penicillin, cefixime, ciprofloxacin and ceftriaxone; 36 isolates of the second group were resistant to tetracycline, ciprofloxacin and penicillin; 17 isolates of the third group were resistant to ciprofloxacin and showed different types of resistance to penicillin and tetracycline; 7 isolates of the forth group were resistant to tetracycline and ciprofloxacin, and only 5 specimens were positive for the beta-lactamase test; 12 isolates of the fifth group were resistant to ciprofloxacin and penicillin.

### 3.3 MALDI-TOF peptide barcoding of antibiotic resistant *N. gonorrhoeae* isolates

A collection of 93 antibiotic-resistant *N. gonorrhoeae* independent isolates was investigated to validate the typing scheme by MALDI-TOF MS. Peptide masses between 2–20 kDa were collected and analyzed using ClinProTool version 3.0 software (Bruker Daltonik, GmbH). All of the peptides delivered an adequate number and intensity (more than $10^4$ a.u.) of peptide

masses. All isolates used in this study were confirmed as *N. gonorrhoeae* by the identification scores obtained from BioTyper analysis. To determine whether peptide barcode could be used to distinguish resistance to different antibiotics of *N. gonorrhoeae*, the total of single MALDI-TOF mass spectra of 93 isolates were subjected to unsupervised hierarchical clustering analyses using ClinProTools. Peptides from each individual cluster generated particular peptide barcode that contained their individual unique mass(es) and differentiated each antibiotic-resistance from one another (Fig 1A and 1B). Hierarchical cluster analysis of the 93 isolates by ClinProTools revealed the different peptidome profiles from 2 to 20 kDa among the 5 groups: AZ, C, CP, CT and CPT (Fig 1A). Therefore, there are differentially expressed peptide profiles

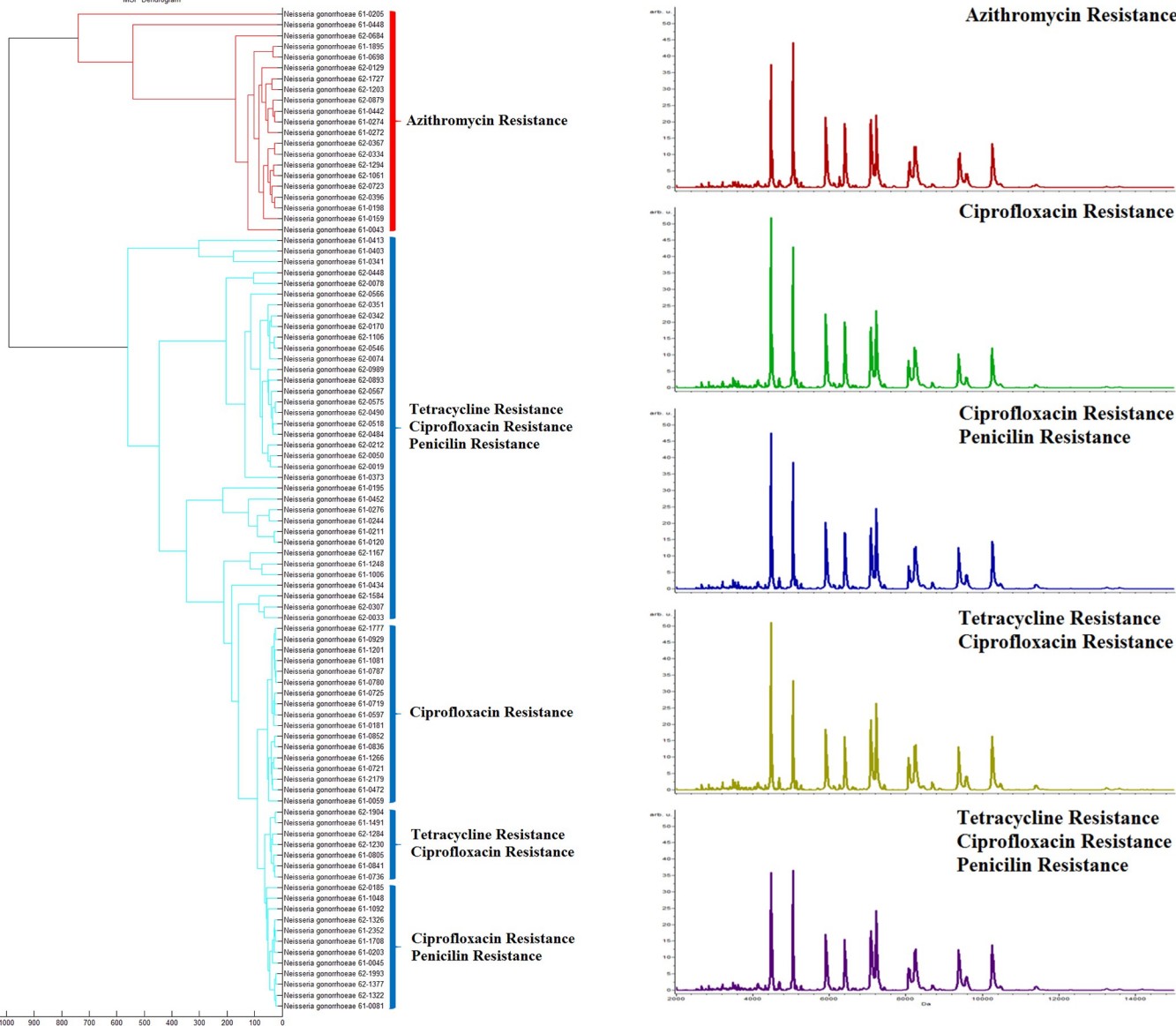

**Fig 1. Dendrogram and MALDI-TOF peptide barcodes of 93 multidrug resistant *N. gonorrhoeae* clinical isolates.** (A) Dendrogram of Hierarchical Cluster Analysis based on the peptide barcodes. (B) Average MALDI-TOF peptide barcodes of azithromycin resistance (AZ), ciprofloxacin resistance (C), ciprofloxacin and penicillin resistance (CP), ciprofloxacin and tetracycline resistance (CT), ciprofloxacin, penicillin and tetracycline resistance (CPT). The X-axis represents the mass to charge ratio (m/z) and the Y-axis represents the intensity of the spectra.

in all 5 groups: AZ (n = 21) was resistant to azithromycin with MIC at 1–4 μg/mL; CPT (n = 36) were resistant to ciprofloxacin, penicillin and tetracycline with MICs at 2–4, 16–64 and 8–32 μg/mL, respectively; C (n = 17) was resistant to ciprofloxacin with MIC of 1–32 μg/mL; CT (n = 7) was resistant to ciprofloxacin and tetracycline with MIC at 1–16 and 16–32 μg/mL, repectively; and CP (n = 12) was resistant to ciprofloxacin and penicillin with MICs at 1–16 and 4–64 μg/mL, respectively. *Dendrogram* of Hierarchical Cluster Analysis revealed that C, CP, CT and CPT were clustered in the same clade while a clade of *AZ* isolates showed a greater distribution from the others (Fig 1A). In addition, the *CP* and *CT* isolates were grouped closer and displayed the shortest distance among all the bacterial isolates tested. This indicated a high similarity in peptidome profiles between *CP* and *CT*. Based on this study, the peptide barcodes generated by MALDI-TOF MS could be used to distinguish each antibiotic-resistance from one another. The average spectrum of each *group* was generated by a combination of raw MALDI-TOF mass spectra using the ClinProTools software (Fig 1B).

### 3.4 LC-MS analysis of antibiotic resistant *N. gonorrhoeae* isolates

Previous studies have reported the detection of antibiotic resistance bacteria based on peptide barcode differences [18–21]. However, the specific antibiotic resistance peptides on MALDI-TOF mass spectra are difficult to identify due to their low abundance or high molecular weights. LC-MS was therefore used to improve the identification and quantification of potential peptide biomarkers with greater dynamic range, higher resolution, reproducibility and accuracy [22]. The peptidome samples of each antibiotic resistant *N. gonorrhoeae* including AZ, C, CP, CT, CPT were pooled and the differential expression of peptides in each group were determined using LC-MS. All 1,828 peptides were significantly identified using the DeCyder MS Differential Analysis software (p < 0.05). A total of 1,795, 1,758, 1,711, 1,691, and 1,773 peptides were identified in the peptidomes of AZ, C, CP, CT, CPT, respectively (Fig 2). Principal component analysis (PCA) was then applied to explain the differentiation between peptidomes from AZ, C, CP, CT and CPT (Fig 3). The low dimensional variations (PC1 11.3% and PC2 8.9%) suggested that most peptides were almost identical among all groups.

An ANOVA was performed to determine which peptides were changed in each peptidome. Fisher's LSD were performed for all post hoc tests. There were 9 peptides significantly differentially overexpressed among all analyzed groups ($p < 0.05$) as shown in Table 1. They were then analyzed for their intersections among the different sample groups using venn diagram as shown in Fig 4. VNCNPETV peptide of carbamoyl-phosphate synthase large chain (carB) was markedly detected in all groups. In particular, 8 peptides were identified in C, while 8 peptides were identified in CP. Four peptides were found in CT. Eight peptides were identified in CPT. Overall, 9 peptides derived from AZ, C, CP, CT, CPT proteins with known functions, matched in the STITCH database, were selected and predicted their association with antibiotics including azithromycin, ciprofloxacin, penicillin and tetracycline as shown in Fig 5.

## 4. Discussion

Antibiotic-resistant gonorrhea has been a chronic public health burden in many countries in the world. Since 2020, WHO reports that there are increasing newly infected cases among adolescents and adults worldwide. MALDI-TOF MS has been recently used to discover several specific peptide patterns of clinical microorganism [23, 24]. In the present study, peptide barcodes and peptidomic clusters were obtained from 93 clinical isolates of multidrug resistant *N. gonorrhoeae*. Distinct peptide barcodes among AZ, C, CP, CT, and CPT were observed. The MALDI-TOF mass spectra exhibited good qualities, a high intensity, and an adequate number

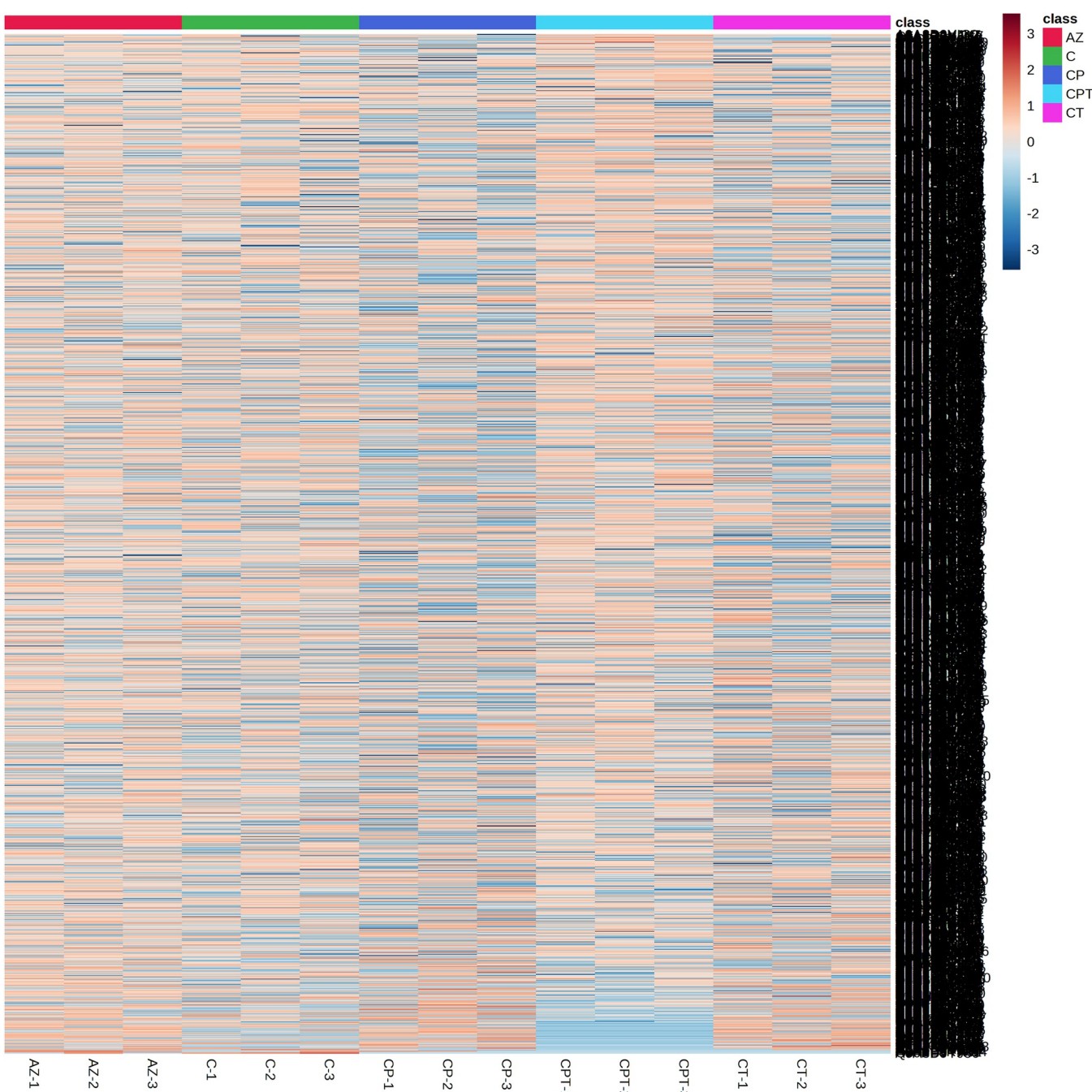

**Fig 2. A heatmap of differentially expressed peptides in multidrug resistant *N. gonorrhoeae*.** Peptides expressed differentially in azithromycin resistance (AZ), ciprofloxacin resistance (C), ciprofloxacin and penicillin resistance (CP), ciprofloxacin and tetracycline resistance (CT), ciprofloxacin, penicillin and tetracycline resistance (CPT). The peptidome analysis was done in triplicate.

of peptide masses that overall indicated that this method could be applicable for multidrug resistant *N. gonorrhoeae* classification.

The peptide barcodes resulted from MALDI-TOF MS were consistent with the data from antimicrobial susceptibility test for differentiation of multidrug resistant *N. gonorrhoeae*. AZ with MIC of azithromycin at concentration of 1–4 μg/mL was defined to be resistance, a

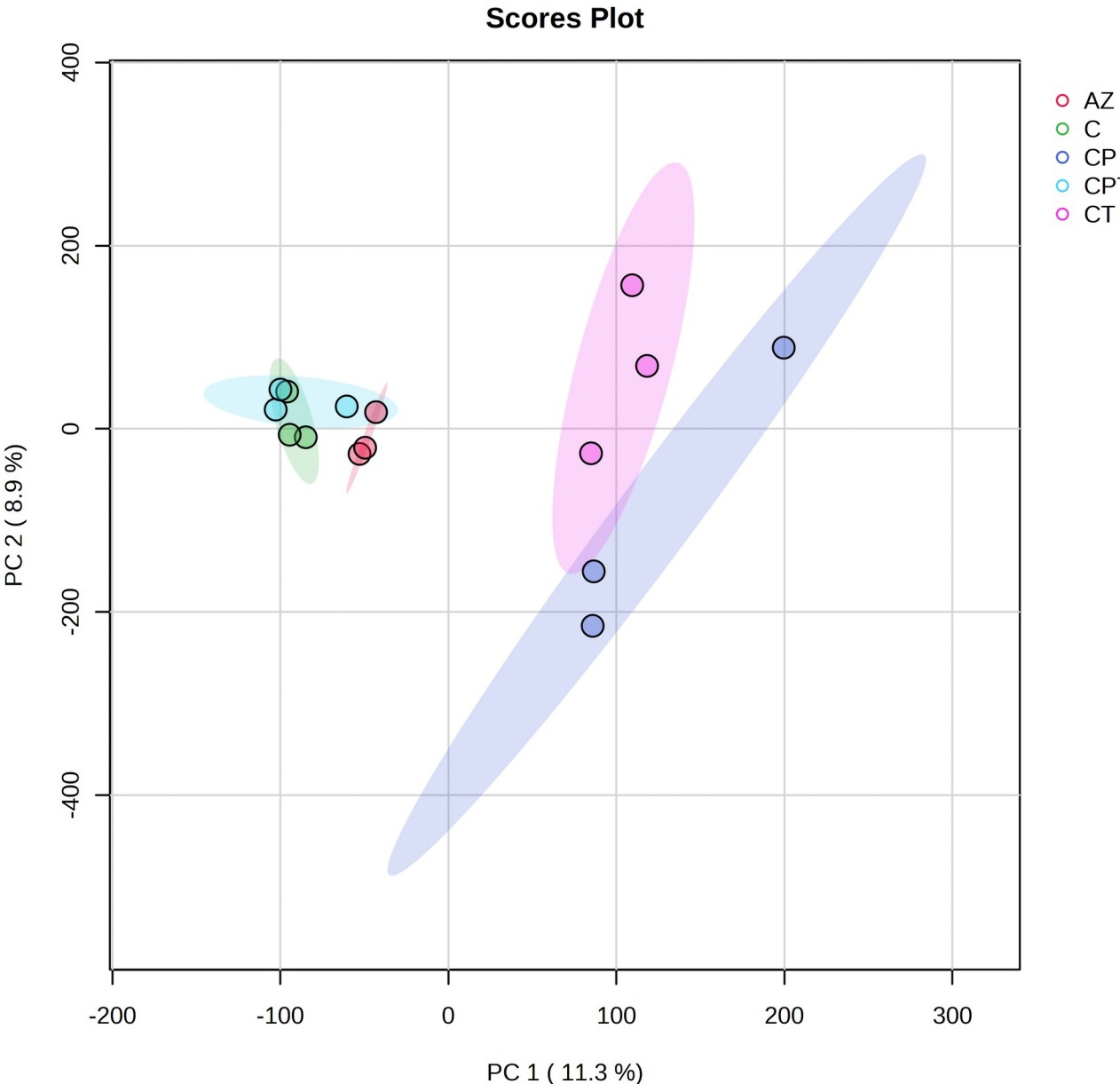

**Fig 3. A principal component analysis (PCA) of multidrug resistant *N. gonorrhoeae* peptidome.** Separation of peptide profiles of azithromycin resistance (AZ), ciprofloxacin resistance (C), ciprofloxacin and penicillin resistance (CP), ciprofloxacin and tetracycline resistance (CT), ciprofloxacin, penicillin and tetracycline resistance (CPT). The peptidome analysis was done in triplicate.

finding similar to the result from MALDI-TOF MS. The data from this study were similar to the previous report by Pham et al. (2020) which differentiated *N. gonorrhoeae* with different levels of susceptibility to azithromycin into two subpopulations by using MALDI-TOF MS [25]. Therefore, the peptide barcodes generated by MALDI-TOF MS could be considered as a high throughput and high sensitivity diagnostic tool for rapid screening of multidrug resistant *N. gonorrhoeae*. Since the accuracy of the classification depends on the number of reference spectra present in the database [26], for further study, more peptide barcodes from clinical isolates of multidrug resistant *N. gonorrhoeae* should be included in the database to improve the reliability of the classification.

**Table 1. List of identified 9 peptides significantly expressed in AZ, C, CP, CT, and CPT.**

| Protein | Protein name | Peptide sequence | Function |
|---|---|---|---|
| NLA_15390 | Amino acid permease ATP-binding protein | AKQFLQ | Transport of xenobiotics |
| carB | Carbamoyl-phosphate synthase large chain | VNCNPETV | Biosynthesis of amino acid |
| pgk | Phosphoglycerate kinase | GTAHRAQ | Carbon metabolism, biosynthesis of amino acid, glycolysis and gluconeogenesis, microbial metabolism in diverse environments, biosynthesis of secondary metabolites |
| PilC | PilC protein | AMAFYL | Assembly and adherence of gonococcal pili |
| ppk | Polyphosphate kinase | DPAVLAVKM | Growth and pathogenicity |
| proC | Pyrroline-5-carboxylate reductase | FDMAE | Biosynthesis of amino acid, biosynthesis of secondary metabolites |
| NLA_12880 | T cell/B cell stimulating protein TspB | TYGCYGV | Protection against host immune response |
| tbpB | Transferrin-binding protein B | TFTIDAMI | Iron transport |
| tkt | Transketolase | MAALMKI | Carbon metabolism, biosynthesis of amino acid, microbial metabolism in diverse environments, biosynthesis of secondary metabolites |

Although the application of MALDI-TOF MS for identification of microorganisms has been successful, the single laser MALDI approach produces low ion yield resulting in missing low abundant or hardly ionizable molecules [27]. The low-resolution associated with the linear TOF analyzers and the related mass accuracy might limit the discovery of peptide biomarkers. In addition, MALDI-TOF MS does not allow de novo peptide sequencing in terms of resolution and ability to perform peptide fragmentation and hence identify those specific peptide peaks. According to the limitations of MALDI-TOF MS, mass spectra linked directly to specific antibiotic resistance peptides are difficult to identify. LC-MS was therefore used to identify and quantify potential antimicrobial resistance peptides.

Peptidomics analysis by LC-MS showed that nine peptides derived from amino acid permease ATP-binding protein (NLA_15390), carbamoyl-phosphate synthase large chain (carB), phosphoglycerate kinase (pgk), PilC protein (PilC), polyphosphate kinase (ppk), pyrroline-5-carboxylate reductase (proC), T cell/B cell stimulating protein TspB (NLA_12880), transferrin-binding protein B (tbpB), and transketolase (tkt) were significantly identified. VNCNPETV peptide of carbamoyl-phosphate synthase large chain (carB) was markedly increased in AZ, C, CP, CT, and CPT. FDMAE peptide of pyrroline-5-carboxylate reductase, AKQFLQ peptide of amino acid permease ATP-binding protein, AMAFYL peptide of PilC protein, TFTIDAMI peptide of transferrin-binding protein B, GTAHRAQ peptide of phosphoglycerate kinase, DPAVLAVKM peptide of polyphosphate kinase and MAALMKI peptide of transketolase were detected only in C and related CP, CT and CPT (Fig 4). These 9 peptides could be used to distinguish AZ, C, CP, CT, and CPT clusters.

Amino acid permease ATP-binding protein is utilized to evade the toxic effects of antibiotics from the *Neisseria* cells [28]. Carbamyl phosphate synthetase helps gonococci to utilize ornithine for growth in place of citrulline [29]. PilC, a factor known to function in the assembly and adherence of gonococcal pili, are essential for transformation competence [30]. Polyphosphate kinase involved in the accumulation of polyphosphate which played an important role in growth and pathogenicity of gonococci [31]. TspB mediated IgG binding and aggregate/biofilm formation which may provide protection against immune responses [32]. Transferrin binding protein B (*tbpB*), an outer membrane lipoprotein, is required for the acquisition of iron from human transferrin [33]. It is considered to be potentially important vaccine and therapeutic targets [34, 35].

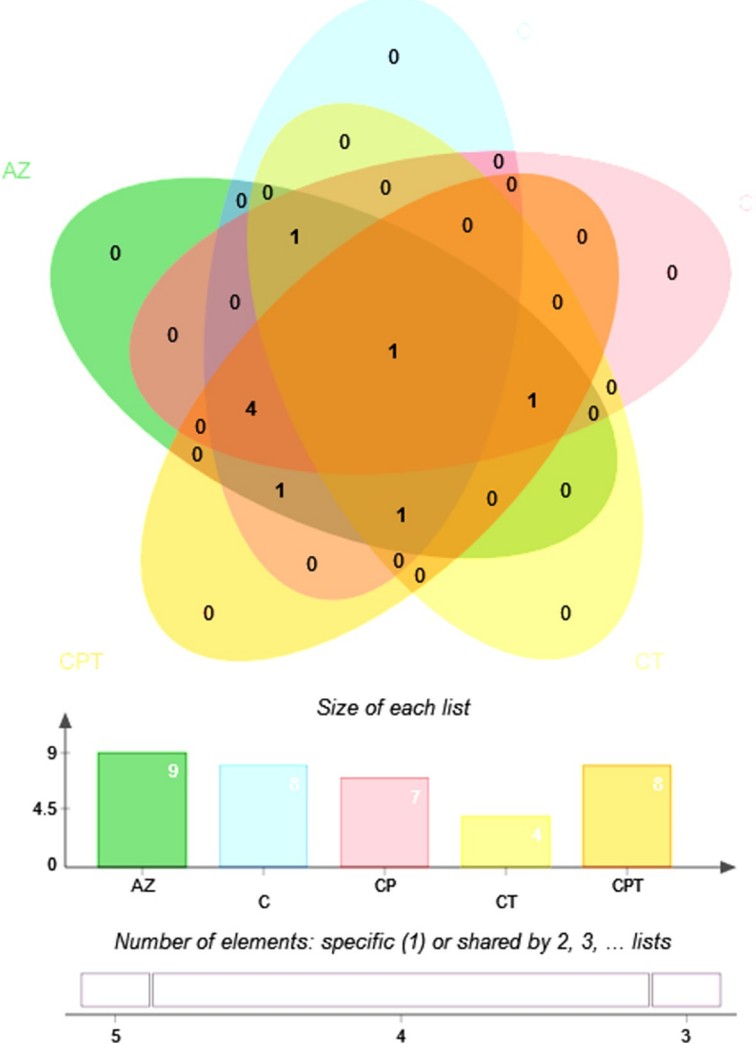

**Fig 4. Distribution of peptides identified in multidrug resistant *N. gonorrhoeae*.** A Venn diagram of 9 peptides significantly expressed in azithromycin resistance (AZ), ciprofloxacin resistance (C), ciprofloxacin and penicillin resistance (CP), ciprofloxacin and tetracycline resistance (CT), ciprofloxacin, penicillin and tetracycline resistance (CPT).

The protein interactions network of amino acid permease ATP-binding protein (NLA_15390), carbamoyl-phosphate synthase large chain (carB), phosphoglycerate kinase (pgk), pyrroline-5-carboxylate reductase (proC), T cell/B cell stimulating protein TspB (NLA_12880), and transketolase (tkt) showed the strong relationship with antibiotic resistances, including azithromycin, penicillin, tetracycline and ciprofloxacin using STITCH 5.0 (Fig 5). These proteins have been reported to play a crucial role in numerous pathways as shown in Kyoto Encyclopedia of Genes and Genomes (KEGG) pathway maps, namely glycolysis/gluconeogenesis, biosynthesis of amino acid, carbon metabolism, microbial metabolism in diverse environments and biosynthesis of secondary metabolites [36].

Peptide biomarkers of antibiotic resistance were associated with resistance enzymes including beta-lactamases, kanR and aminoglycoside modifying enzyme [20]. Some of these resistance peptides were encoded by gene cassettes in the bacterial genomes [21]. However, most of the bacterial peptides were generated by protein degradation [37]. Nine unique peptides were

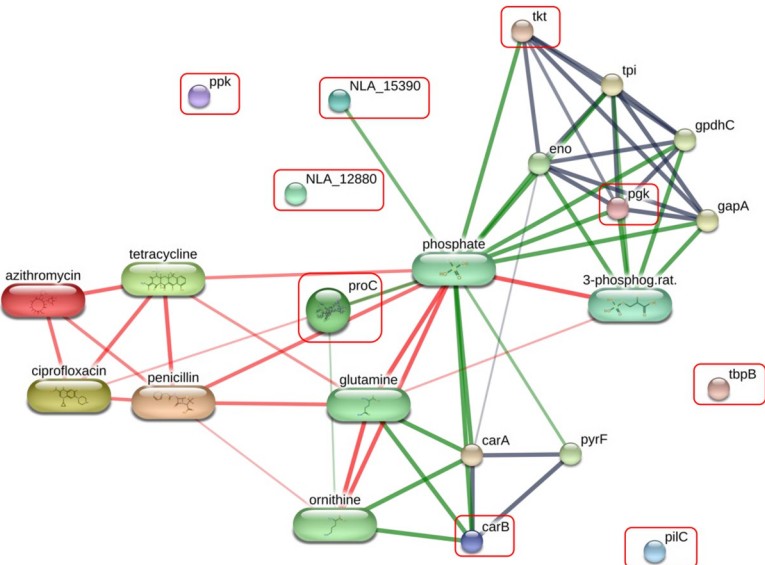

**Fig 5. Association of peptides identified in multidrug resistant *N. gonorrhoeae*.** Involvements of amino acid permease ATP-binding protein (NLA_15390), carbamoyl-phosphate synthase large chain (carB), phosphoglycerate kinase (pgk), PilC protein (PilC), polyphosphate kinase (ppk), pyrroline-5-carboxylate reductase (proC), transferrin-binding protein B (tbpB), and transketolase (tkt) in networks of protein–antibiotic agent interactions: azithromycin, ciprofloxacin, penicillin and tetracycline.

detected in each antibiotic resistant group that originated from nine proteins. These peptides could serve as specific biomarker candidates for AZ, C, CP, CT and CPT-resistant *N. gonorrhoeae*. This could probably lead to the development of whole cell biomarker(s) for efficient diagnosis and drug development. This might be of great benefit for medical researchers to understand the pathogenesis and biological process of multidrug resistant *N. gonorrhoeae*.

## 5. Conclusions

MALDI-TOF MS has potential to be used for rapid screening multidrug resistant *N. gonorrhoeae*. With the combination of peptide barcodes (generated by MALDI-TOF MS) and peptidome analysis (by LC MS), potential peptide candidates associated with the multidrug resistant *N. gonorrhoeae* were identified. Nine peptides, NLA_15390, carB, pgk, PilC, ppk, proC, NLA_12880, tbpB, and tkt, were identified and might be used as potential biomarkers for diagnosis of the disease. The interaction network between peptides and antibiotics was observed. Further studies should be performed in a larger population to evaluate these potential candidate biomarkers.

## Acknowledgments

We would like to thank staff members of Sexually Transmitted Infection Cluster, Bureau of AIDS TB and STIs, Department of Disease Control, Ministry of Public Health, Thailand, for kind supports in providing technical materials and clinical strains of multidrug resistant *N. gonorrheoae*.

## Author Contributions

**Conceptualization:** Sittiruk Roytrakul, Teerakul Arpornsuwan.

**Data curation:** Sittiruk Roytrakul, Pongsathorn Sangprasert, Janthima Jaresitthikunchai, Narumon Phaonakrop, Teerakul Arpornsuwan.

**Formal analysis:** Sittiruk Roytrakul, Pongsathorn Sangprasert, Janthima Jaresitthikunchai, Narumon Phaonakrop, Teerakul Arpornsuwan.

**Funding acquisition:** Teerakul Arpornsuwan.

**Investigation:** Pongsathorn Sangprasert, Janthima Jaresitthikunchai, Narumon Phaonakrop.

**Methodology:** Sittiruk Roytrakul, Pongsathorn Sangprasert, Janthima Jaresitthikunchai, Narumon Phaonakrop, Teerakul Arpornsuwan.

**Visualization:** Sittiruk Roytrakul.

**Writing – original draft:** Teerakul Arpornsuwan.

**Writing – review & editing:** Sittiruk Roytrakul.

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
