## [Decision Letter · Decision Letter 0]

25 Apr 2023

PONE-D-23-07664Peptide barcode of multi drug resistant strains of Neisseria gonorrhoeae isolated from patients in ThailandPLOS ONE

Dear Dr. Arpornsuwan

Thank you for submitting your manuscript to PLOS ONE. After careful consideration, we feel that it has merit but does not fully meet PLOS ONE’s publication criteria as it currently stands. Therefore, we invite you to submit a revised version of the manuscript that addresses the points raised during the review process.

We look forward to receiving your revised manuscript.

Kind regards,

Benjamin M. Liu, MBBS, PhD, D(ABMM), MB(ASCP)

Academic Editor

PLOS ONE

Journal Requirements:

6. Please upload a new copy of Figures 1 and 2 as the detail is not clear. Please follow the link for more information: " ext-link-type="uri" xlink:type="simple">https://blogs.plos.org/plos/2019/06/looking-good-tips-for-creating-your-plos-figures-graphics/"
" ext-link-type="uri" xlink:type="simple">https://blogs.plos.org/plos/2019/06/looking-good-tips-for-creating-your-plos-figures-graphics/"

Reviewers' comments:

Reviewer's Responses to Questions

**Comments to the Author**

1. Is the manuscript technically sound, and do the data support the conclusions?

Reviewer #1: Partly

Reviewer #2: Partly

Reviewer #3: Partly

Reviewer #4: Yes

2. Has the statistical analysis been performed appropriately and rigorously? 

Reviewer #1: No

Reviewer #2: Yes

Reviewer #3: N/A

Reviewer #4: Yes

3. Have the authors made all data underlying the findings in their manuscript fully available?

Reviewer #1: No

Reviewer #2: Yes

Reviewer #3: No

Reviewer #4: Yes

4. Is the manuscript presented in an intelligible fashion and written in standard English?

Reviewer #1: No

Reviewer #2: Yes

Reviewer #3: Yes

Reviewer #4: No

5. Review Comments to the Author

Reviewer #1: Summary : It is a very interesting and relevant topic. Determining the antibiotic susceptibility of Neisseria gonorrhoeae by mass spectrometry(MS) will help in easy and early detection of antibiotic resistance and hence efficient treatment of patients. Identification of biomarkers of resistance is also very useful.

Over all Impression: The study combines two research questions. First, generating the peptide barcodes for the different groups of antibiotic resistant Neisseria gonorrhoeae, secondly discovering the peptide biomarker of the antibiotic resistance by LCMS. The results, data and statistical analysis of both the experiments have not been presented adequately in the manuscript.

Major issues:

1. The result part of the MALDI-TOF peptide barcoding , from line number 203, does not clearly provide clear information and data regarding the differences in the peaks of the spectra of the different antibiotic resistance groups mentioned in the study. The Figure 1 is not clear and does not mention the above information. The statistical analysis and Odd's ratio for the same is not reported.

2. Which test of statistical analysis for the LCMS test has been done? The figures of the of the LCMS tests are not very clear.

Minor issues: I suggest that the authors hire a copy editor.

The manuscript deals with important topics of detection of antibiotic resistance by MALDITOF and discovering peptide biomarker of antibiotic resistance by LCMS. If the authors can provide the aforementioned details, it will be a valuable contribution to the research in antibiotic resistance.

Reviewer #2: 1. Novelty statement should be included, as identified 9 peptides are already known for their function in N. gonorrhoeae.

2. What are the Inclusion and Exclusion criteria?

3. What are the limitations of the study?

4. Why sample size is small?

-------------

Reviewer #3: 1. In the abstract and entire document, the name "Neisseria gonorrhoea" should all be typed in italics

2. Line 39, insert "of" between prevalence and infection.

3. line 56 remove hyphen from the word "de-creased"

4. Line 70, include year for the citation mentioned.

5. Line 96, the initial sentence can be re-written as "Sixty-three of the 91 isolates showed resistance to ........"

6. Line 112 remove hyphen from the word "performed"

7. Change "E test" to "E-test"

8. In section 2.5, I think is will be appropriate to describe or provide more detail on how the different bacterial groups were pooled together.

9. Line 300, use the word "consistent"

10. IMPORTANT COMMENT

There appears to be a significant missing link in the results presented in this study. The results obtained in the MIC determination using the E-test strip is not shown. The authors should present the result and further correlate with the peptidome obtained for the bacterial isolates.

This will provide a further validation of the use of the MALDI-TOF as suggested by the authors.

Reviewer #4: 1. There are some language and grammatical errors throughout the manuscript text that should be corrected.

2. Please define the abbreviations at the first mention of them in the text for example, IRB in line 93; PPNG: in line 97.

3. The data of Table 1 is repeated in the text, so this table could be deleted.

6. PLOS authors have the option to publish the peer review history of their article (what does this mean?). If published, this will include your full peer review and any attached files.

Reviewer #1: No

Reviewer #2: No

Reviewer #3: **Yes: **Dr. Ayodele Oluwaseun AJAYI

Reviewer #4: **Yes: **Mohammad Hossein Ahmadi

---

## [Author Response · Author response to Decision Letter 0]

9 Jun 2023

PONE-D-23-07664

Peptide barcode of multi drug resistant strains of Neisseria gonorrhoeae isolated from patients in Thailand

Reviewer #1: Summary: It is a very interesting and relevant topic. Determining the antibiotic susceptibility of Neisseria gonorrhoeae by mass spectrometry (MS) will help in easy and early detection of antibiotic resistance and hence efficient treatment of patients. Identification of biomarkers of resistance is also very useful.

Overall Impression: The study combines two research questions. First, generating the peptide barcodes for the different groups of antibiotic resistant Neisseria gonorrhoeae, secondly discovering the peptide biomarker of the antibiotic resistance by LC-MS. The results, data and statistical analysis of both the experiments have not been presented adequately in the manuscript.

Major issues:

1. The result part of the MALDI-TOF peptide barcoding, from line number 203, does not clearly provide clear information and data regarding the differences in the peaks of the spectra of the different antibiotic resistance groups mentioned in the study. The Figure 1 is not clear and does not mention the above information. The statistical analysis and Odd's ratio for the same is not reported.

Ans. More informations regarding differences in the Maldi-TOF mass spectra of the different antibiotic resistance groups (Figure 1) and statistical analysis were added in Material and Method section (Line 135-142) and Result section (Line 222-230; Line 236-243). 

2. Which test of statistical analysis for the LC-MS test has been done? The figures of the LC-MS tests are not very clear.

Ans. More informations regarding statistical analysis for the LC-MS data by DecyderMS and Mascot were added in Material and Method section (Line 167-174; Line 179-181) and Result section (Line 254-259; Line 261-262). In addition, the peptidomic heatmap, principal component analysis (PCA) and analysis of variance (ANOVA) analysis generated by Metaboanalyst 5.0 were added into Material and Method section (Line 179-181) and Result section (Line 261-262; Line 264-265; Line 267-268). 

3. Minor issues: I suggest that the authors hire a copy editor.

Ans. This manuscript was corrected by The Professional editor under the program of Research clinic that supported by Faculty of Allied Health Sciences Thammasat University.

4. The manuscript deals with important topics of detection of antibiotic resistance by Maldi-TOF and discovering peptide biomarker of antibiotic resistance by LC-MS. If the authors can provide the aforementioned details, it will be a valuable contribution to the research in antibiotic resistance.

Ans. As mentioned in Line 254-259. Previous studies have reported the detection of antibiotic resistance bacteria based on peptide barcode differences [18-21]. However, the specific antibiotic resistance peptides on MALDI-TOF mass spectra are difficult to identify due to their low abundance or high molecular weights. LC-MS/MS was therefore used to improve the identification and quantification of potential peptide biomarkers with greater dynamic range, higher resolution, reproducibility and accuracy [23].

Reviewer #2: 

1. Novelty statement should be included, as identified 9 peptides are already known for their function in N. gonorrhoeae.

Ans. The statement is added (Line 351-357). Peptide biomarkers of antibiotic resistance were associated with resistance enzymes including beta-lactamases, kanR and aminoglycoside modifying enzyme [20]. Some of these resistance peptides were encoded by gene cassettes in the bacterial genomes [21]. However, most of the bacterial peptides were generated by protein degradation [36]. Nine unique peptides were detected in each antibiotic resistant group that originated from nine proteins. These peptides could serve as specific biomarker candidates for AZ, C, CP, CT and CPT-resistant N. gonorrhoeae.

2. What are the Inclusion and Exclusion criteria?

Ans. There is no Inclusion and Exclusion criteria for collecting multidrug resistant strains of Neisseria gonorrhoeae. We used the antibiotic-resistant N. gonorrhoeae samples collected by the Sexually Transmitted Infection Cluster, Bureau of AIDS TB and STIs, Department of Disease Control, Ministry of Public Health, Thailand since January 2019 to December 2021. 

3. What are the limitations of the study?

Ans. The limitation of the study is sample size. Although the the collecting time was 2 years (January 2019 to December 2021), only 93 samples were obtained.

4. Why sample size is small?

Ans. Sample size is a limitation of this study. As shown in Material and Method section, only 93 samples were collected during January 2019 to December 2021.

-------------

 

Reviewer #3: 

1. In the abstract and entire document, the name "Neisseria gonorrhoea" should all be typed in italics

Ans. The name "Neisseria gonorrhoea" is now typed in italics both in the abstract and entire document.

2. Line 39, insert "of" between prevalence and infection.

Ans. "of" is inserted between prevalence and infection

3. line 56 remove hyphen from the word "de-creased"

Ans. hyphen from the word "de-creased" is removed.

4. Line 70, include year for the citation mentioned.

Ans. The year “2014” is added for the citation mentioned. 

5. Line 96, the initial sentence can be re-written as "Sixty-three of the 91 isolates showed resistance to ........"

Ans. This sentence is re-written.

6. Line 112 remove hyphen from the word "performed"

Ans. Hyphen was removed from the word "performed"

7. Change "E test" to "E-test"

Ans. "E test" were changed to "E-test" through the manuscript.

8. In section 2.5, I think is will be appropriate to describe or provide more detail on how the different bacterial groups were pooled together.

Ans. More detail on how the different bacterial groups were pooled together was described in section 2.5 (Line 146-154). The bacterial peptides were extracted from the N. gonorrhoeae according to a previous study with slightly modification [11]. Briefly, 5% (v/v) trifluoroacetic acid (TFA) in absolute acetonitrile (ACN) was added to the culture and the suspension was dissolved by gentle vortexing. The samples were dried and resuspended with 0.1% (v/v) formic acid. The Lowry assay was used to determine peptide concentration [12]. Peptides were equally pooled from each bacterium within the same group. Peptidome samples of each bacterial group including azithromycin resistance group (AZ), ciprofloxacin resistance group (C), ciprofloxacin and penicillin resistance group (CP), ciprofloxacin and tetracycline resistance group (CT), ciprofloxacin, penicillin and tetracycline resistance group (CPT) were analyzed by LC-MS. 

9. Line 300, use the word "consistent"

Ans. Line 300 is corrected by the Professional editor under the program of Research clinic that supported by Faculty of Allied Health Sciences Thammasat University. 

10. IMPORTANT COMMENT: There appears to be a significant missing link in the results presented in this study. The results obtained in the MIC determination using the E-test strip is not shown. The authors should present the result and further correlate with the peptidome obtained for the bacterial isolates. This will provide a further validation of the use of the MALDI-TOF as suggested by the authors.

Ans. The correlation between MICs and peptide barcode is not performed because of broad range of MICs of each bacterial group especially ciprofloxacin and penicillin resistance group (CP), ciprofloxacin and tetracycline resistance group (CT), ciprofloxacin, penicillin and tetracycline resistance group (CPT). Therefore, this study showed that MALDI-TOF peptide barcoding can be used to differentiate between AZ, C, CP, CT and CPT. In addition, LC-MS is used to quantitate and identify potential peptide candidates associated with the multidrug resistant N. gonorrhoeae. 

Reviewer #4: 

1. There are some language and grammatical errors throughout the manuscript text that should be corrected.

Ans. This manuscript was corrected by The Professional editor under the program of Research clinic that supported by Faculty of Allied Health Sciences Thammasat University. 

2. Please define the abbreviations at the first mention of them in the text for example, IRB in line 93; PPNG: in line 97.

Ans. Institutional Review Board (IRB) (Line 94) and penicillinase-producing Neisseria gonorrhoeae (PPNG) were added (Line 96-97). 

3. The data of Table 1 is repeated in the text, so this table could be deleted.

Ans. Table 1 is deleted as suggested.

---

## [Decision Letter · Decision Letter 1]

6 Jul 2023

PONE-D-23-07664R1Peptide barcode of multi drug resistant strains of Neisseria gonorrhoeae isolated from patients in ThailandPLOS ONE

Dear Dr. Arpornsuwan,

Thank you for submitting your manuscript to PLOS ONE. After careful consideration, we feel that it has merit but does not fully meet PLOS ONE’s publication criteria as it currently stands. Therefore, we invite you to submit a revised version of the manuscript that addresses the points raised during the review process.

We look forward to receiving your revised manuscript.

Kind regards,

Benjamin M. Liu, MBBS, PhD, D(ABMM), MB(ASCP)

Academic Editor

PLOS ONE

Journal Requirements:

Reviewers' comments:

Reviewer's Responses to Questions

**Comments to the Author**

1. If the authors have adequately addressed your comments raised in a previous round of review and you feel that this manuscript is now acceptable for publication, you may indicate that here to bypass the “Comments to the Author” section, enter your conflict of interest statement in the “Confidential to Editor” section, and submit your "Accept" recommendation.

Reviewer #1: All comments have been addressed

Reviewer #2: All comments have been addressed

Reviewer #4: All comments have been addressed

2. Is the manuscript technically sound, and do the data support the conclusions?

Reviewer #1: Yes

Reviewer #2: Yes

Reviewer #4: Yes

3. Has the statistical analysis been performed appropriately and rigorously? 

Reviewer #1: Yes

Reviewer #2: Yes

Reviewer #4: Yes

4. Have the authors made all data underlying the findings in their manuscript fully available?

Reviewer #1: Yes

Reviewer #2: Yes

Reviewer #4: Yes

5. Is the manuscript presented in an intelligible fashion and written in standard English?

Reviewer #1: Yes

Reviewer #2: Yes

Reviewer #4: Yes

6. Review Comments to the Author

Reviewer #1: Summary:The issues raised in the previous review have been addressed adequately. The manuscript is also presented in intelligible fashion.

Minor issue:

1.The authors could mention the limitations of the study, the MALDI TOF MS in particular as mentioned in the response to reviewers "However, the specific antibiotic resistance peptides on MALDI-TOF mass spectra are difficult to identify due to their low abundance or high molecular weights"

Reviewer #2: (No Response)

Reviewer #4: The authors have adequately addressed all my comments raised in the previous round of review and I think this manuscript is now acceptable for publication.

7. PLOS authors have the option to publish the peer review history of their article (what does this mean?). If published, this will include your full peer review and any attached files.

Reviewer #1: No

Reviewer #2: **Yes: **Shivankar Agrawal

Reviewer #4: **Yes: **Mohammad Hossein Ahmadi

---

## [Author Response · Author response to Decision Letter 1]

8 Jul 2023

PONE-D-23-07664R1

Peptide barcode of multidrug resistant strains of Neisseria gonorrhoeae isolated from patients in Thailand 

Journal requirements

1. The reference list in line 509-551 has been changed from no 27-37, The reference number 27 was added in line 322 and in line 509 for reference list. The front color of reference no 26 in line 506 was formatted. All References were reviewed and corrected to be complete. 

27. Karas M., Glückmann M., Schäfer J. Ionization in Matrix-Assisted Laser Desorption/Ionization: Singly Charged Molecular Ions Are the Lucky Survivors. J. Mass Spectrom. 2000; 35:1–12. doi: 10.1002/(SICI)1096-9888(200001)35:11::AID-JMS9043.0.CO;2-0

2. We upload all new copy of Figures following by PACE.

Response to Reviewers

Reviewer #1: Summary: The issues raised in the previous review have been addressed adequately. The manuscript is also presented in intelligible fashion.

Minor issue:

1. The authors could mention the limitations of the study, the MALDI TOF MS in particular as mentioned in the response to reviewers "However, the specific antibiotic resistance peptides on MALDI-TOF mass spectra are difficult to identify due to their low abundance or high molecular weights"

Answer: The sentences regarding the limitations of the study were added in line 320-328. “Although the application of MALDI-TOF MS for identification of microorganisms has been successful, the single laser MALDI approach produces low ion yield resulting in missing low abundant or hardly ionizable molecules [27]. The low-resolution associated with the linear TOF analyzers and the related mass accuracy might limit the discovery of peptide biomarkers. In addition, MALDI-TOF MS does not allow de novo peptide sequencing in terms of resolution and ability to perform peptide fragmentation and hence identify those specific peptide peaks. According to the limitations of MALDI-TOF MS, mass spectra linked directly to specific antibiotic resistance peptides are difficult to identify. LC-MS/MS was therefore used to identify and quantify potential antimicrobial resistance peptides.”

---

## [Editor Report · Decision Letter 2]

17 Jul 2023

Peptide barcode of multi drug resistant strains of Neisseria gonorrhoeae isolated from patients in Thailand

PONE-D-23-07664R2

Dear Dr. Arpornsuwan,

We’re pleased to inform you that your manuscript has been judged scientifically suitable for publication and will be formally accepted for publication once it meets all outstanding technical requirements.

Kind regards,

Benjamin M. Liu, MBBS, PhD, D(ABMM), MB(ASCP)

Academic Editor

PLOS ONE
---

## [Editor Report · Acceptance letter]

27 Jul 2023

PONE-D-23-07664R2 

Peptide barcode of multidrug-resistant strains of *Neisseria gonorrhoeae* isolated from patients in Thailand 

Dear Dr. Arpornsuwan:

I'm pleased to inform you that your manuscript has been deemed suitable for publication in PLOS ONE. Congratulations! Your manuscript is now with our production department. 

Kind regards, 

on behalf of

Dr. Benjamin M. Liu 

Academic Editor

PLOS ONE